# Video Prediction Transformers without Recurrence or Convolution

**Yujin Tang**[*]                                                                *tangyujin0275@gmail.com*
*Shanghai Jiao Tong University*
*University of California, Merced*

**Lu Qi**[†]                                                                        *qqlu1992@gmail.com*
*Wuhan University*

**Xiangtai Li**                                                                  *xiangtai94@gmail.com*
*Nanyang Technological University*

**Chao Ma**                                                                     *chaoma99@gmail.com*
*Shanghai Jiao Tong University*

**Ming-Hsuan Yang**                                                    *minghsuanyang@gmail.com*
*University of California, Merced*

**Reviewed on OpenReview:** *https://openreview.net/forum?id=Afvhu9Id8m*

## Abstract

Video prediction has witnessed the emergence of RNN-based models led by ConvLSTM, and CNN-based models led by SimVP. Following the significant success of ViT, recent works have integrated ViT into both RNN and CNN frameworks, achieving improved performance. While we appreciate these prior approaches, we raise a fundamental question: Is there a simpler yet more effective solution that can eliminate the high computational cost of RNNs while addressing the limited receptive fields and poor generalization of CNNs? How far can it go with a simple pure transformer model for video prediction? In this paper, we propose Pred-Former, a framework entirely based on Gated Transformers. We provide a comprehensive analysis of 3D Attention in the context of video prediction. Extensive experiments demonstrate that PredFormer delivers state-of-the-art performance across four standard benchmarks. The significant improvements in both accuracy and efficiency highlight the potential of PredFormer as a strong baseline for real-world video prediction applications. The source code and trained models are released at `https://github.com/yyyujintang/PredFormer`.

## 1 Introduction

Video Prediction (Wang et al., 2018c; Chang et al., 2021; Gao et al., 2022a), also named as Spatio-Temporal predictive learning (Wang et al., 2017; 2018b; Tan et al., 2023a;b) involves learning spatial and temporal patterns by predicting future frames based on past observations. This capability is essential for various applications, including weather forecasting (Rasp et al., 2020; Pathak et al., 2022; Bi et al., 2023), traffic flow prediction (Fang et al., 2019; Wang et al., 2019), precipitation nowcasting (Shi et al., 2015; Gao et al., 2022b) and human motion forecasting (Zhang et al., 2017b; Wang et al., 2018a).

Despite the success of various video prediction methods, they often struggle to balance computational cost and performance. On the one hand, high-powered recurrent-based methods (Shi et al., 2015; Wang et al.,

---

[1]Project Page: `https://yyyujintang.github.io/predformer-project/` ; [*] Work done while visiting University of California, Merced. [†] Corresponding author.

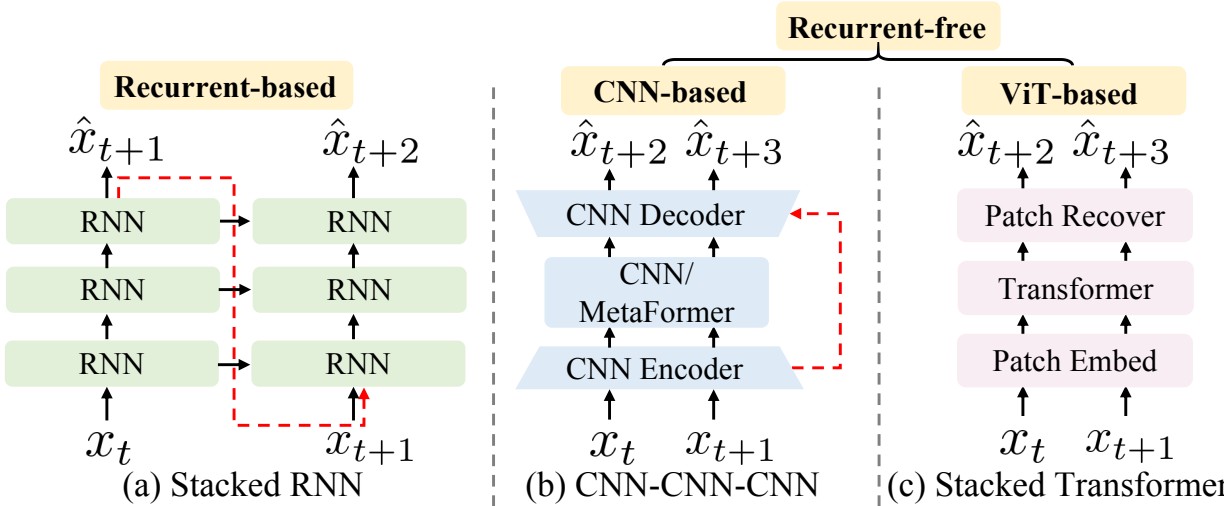

Figure 1: Main categories of video prediction framework. (a) Recurrent-based Framework (b) CNN Encoder-Decoder-based Recurrent-free Framework. (c) Pure transformer-based Recurrent-free Framework.

2017; 2019; Chang et al., 2021; Yu et al., 2019; Tang et al., 2023; 2024) rely heavily on autoregressive RNN frameworks, which face significant limitations in parallelization and computational efficiency. On the other hand, efficient recurrent-free methods (Gao et al., 2022a; Tan et al., 2023a), such as those based on the SimVP framework, use CNNs in an encoder-decoder architecture but are constrained by local receptive fields, limiting their scalability and generalization. The ensuing question is *Can we develop a framework that autonomously learns spatiotemporal dependencies without relying on inductive bias?*

An intuitive solution directly adopts a pure transformer (Vaswani et al., 2017) structure, as it is an efficient alternative to RNNs and has better scalability than CNNs. Transformers have demonstrated remarkable success in visual tasks (Dosovitskiy et al., 2020; Liu et al., 2021; Bertasius et al., 2021; Arnab et al., 2021; Tarasiou et al., 2023). Previous video prediction methods try to combine Swin Transformer (Liu et al., 2021) in recurrent-based frameworks such as SwinLSTM (Tang et al., 2023) and integrate MetaFormer (Yu et al., 2022) as a temporal translator in recurrent-free CNN-based encoder-decoder frameworks such as Open-STL (Tan et al., 2023b). Despite these advances, pure transformer-based architecture remains underexplored mainly due to the challenges of capturing spatial and temporal relationships within a unified framework. While merging spatial and temporal dimensions and applying full attention is conceptually straightforward, it is computationally expensive because of the quadratic scaling of attention with sequence length. Several recent methods (Bertasius et al., 2021; Arnab et al., 2021; Tarasiou et al., 2023) decouple full attention and show that spatial and temporal relations can be treated separately in a factorized or interleaved manner to reduce complexity.

In this work, we propose PredFormer, a pure transformer-based architecture for video prediction. PredFormer dives into the decomposition of spatial and temporal transformers, integrating self-attention with gated linear units (Dauphin et al., 2017) to more effectively capture complex spatiotemporal dynamics. In addition to retaining spatial-temporal full attention encoder and factorized encoder strategies for both spatial-first and temporal-first configurations, we introduce six novel interleaved spatiotemporal transformer architectures, resulting in nine configurations. We explore how far this simple framework can go with different strategies of 3D Attention. This comprehensive investigation pushes the boundaries of current models and sets valuable benchmarks for spatial-temporal modeling.

Notably, PredFormer achieves state-of-the-art performance across four benchmark data sets, including synthetic prediction of moving objects, real-world human motion prediction, traffic flow prediction and weather forecasting, outperforming previous methods by a substantial margin without relying on complex model architectures or specialized loss functions.

The main contributions can be summarized as follows:

- We propose PredFormer, a pure gated transformer-based framework for video prediction. By eliminating the inductive biases inherent in CNNs, PredFormer harnesses the scalability and generalization capabilities of the transformers, achieving significantly enhanced performance ceilings with efficiency.

- We perform an in-depth analysis of spatial-temporal transformer factorization, exploring full-attention encoders and factorized encoders along with interleaved spatiotemporal transformer architectures, resulting in nine PredFormer variants. These variants address the differing spatial and temporal resolutions across tasks and datasets for optimal performance.

- We conduct a comprehensive study on training ViT from scratch on small datasets, exploring regularization and position encoding techniques.

- Extensive experiments demonstrate the state-of-the-art performance of PredFormer. It outperforms SimVP by 48% while achieving $1.5\times$ faster inference speed on Moving MNIST. Besides, PredFormer surpasses SimVP with $8\times$, $5\times$, and $3\times$ inference speed on TaxiBJ, WeatherBench, and Human3.6m, while achieving higher accuracy.

## 2 Related Work

**Recurrent-based video prediction.** Recent advancements in recurrent-based video prediction models have integrated CNNs, ViTs, and Vision Mamba (Liu et al., 2024) into RNNs, employing various strategies to capture spatiotemporal relationships. ConvLSTM (Shi et al., 2015), evolving from FC-LSTM (Srivastava et al., 2015), innovatively integrates convolutional operations into the LSTM framework. PredNet (Lotter et al., 2017) leverages deep recurrent convolutional neural networks with bottom-up and top-down connections to predict future video frames. PredRNN (Wang et al., 2017) introduces the Spatiotemporal LSTM (ST-LSTM) unit, which effectively captures and memorizes spatial and temporal representations by propagating hidden states horizontally and vertically. PredRNN++ (Wang et al., 2018b) incorporates a gradient highway unit and Causal LSTM to address the vanishing gradient problem and adaptively capture temporal dependencies. E3D-LSTM (Wang et al., 2018c) extends the memory capabilities of ST-LSTM by integrating 3D convolutions. The MIM model (Wang et al., 2019) further refines the ST-LSTM by reimagining the forget gate with dual recurrent units and utilizing differential information between hidden states. CrevNet (Yu et al., 2019) employs a CNN-based reversible architecture to decode complex spatiotemporal patterns. PredRNNv2 (Wang et al., 2022) enhances PredRNN by introducing a memory decoupling loss and a curriculum learning strategy. MAU (Chang et al., 2021) adds a motion-aware unit to capture dynamic motion information. SwinLSTM (Tang et al., 2023) integrates the Swin Transformer (Liu et al., 2021) module into the LSTM architecture, while VMRNN (Tang et al., 2024) extends this by incorporating the Vision Mamba module. Unlike these approaches, PredFormer is a recurrent-free method that offers superior efficiency.

**Recurrent-free video prediction.** Recent recurrent-free models, e.g., SimVP (Gao et al., 2022a), are developed based on a CNN-based encoder-decoder with a temporal translator. TAU (Tan et al., 2023a) builds upon this by separating temporal attention into static intra-frame and dynamic inter-frame components, introducing a differential divergence loss to supervise inter-frame variations. OpenSTL (Tan et al., 2023b) integrates a MetaFormer model as the temporal translator. Additionally, PhyDNet (Guen & Thome, 2020) incorporates physical principles into CNN architectures, while DMVFN (Hu et al., 2023) introduces a dynamic multi-scale voxel flow network to enhance video prediction performance. EarthFormer (Gao et al., 2022b) presents a 2D CNN encoder-decoder architecture with cuboid attention.WAST (Nie et al., 2024) proposes a wavelet-based method, coupled with a wavelet-domain High-Frequency Focal Loss. In contrast to prior methods, PredFormer advances video prediction with its recurrent-free, pure transformer-based architecture, leveraging a global receptive field to achieve superior performance, outperforming prior models without relying on complex architecture designs or specialized loss.

Recurrent-based approaches struggle with parallelization and performance, while CNN-based recurrent-free methods often sacrifice scalability and generalization despite their strong inductive biases. In contrast

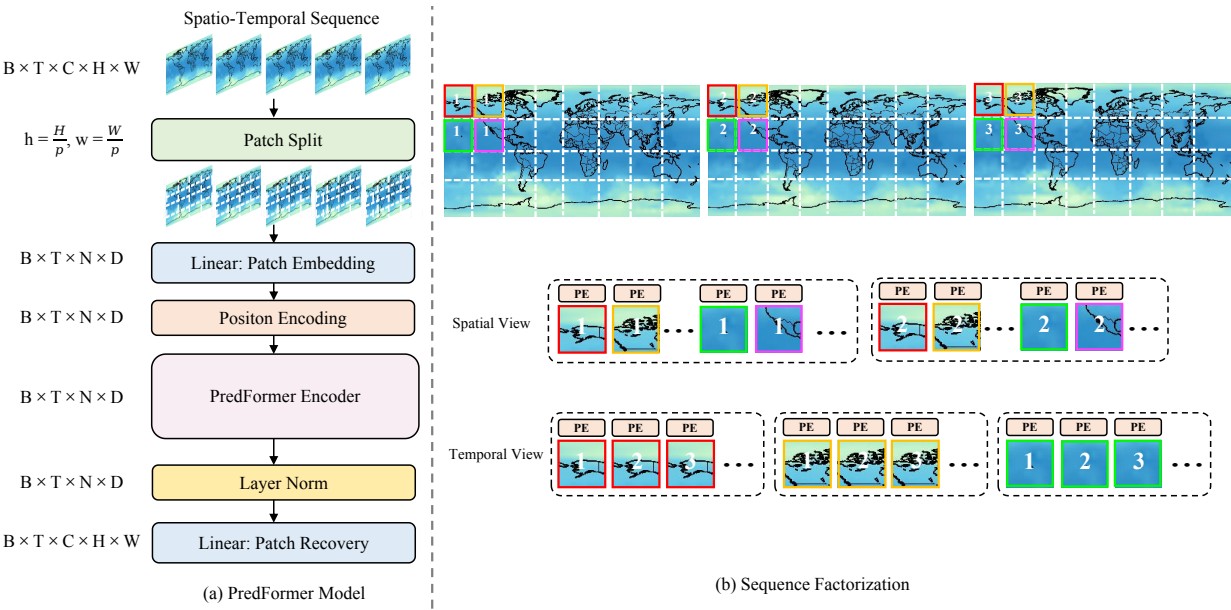

Figure 2: Overview of the PredFormer framework.

to prior methods, PredFormer advances video prediction with its recurrent-free, pure transformer-based architecture, leveraging a global receptive field to achieve superior performance, outperforming prior models without relying on complex model designs or specialized loss designs.

**Vision Transformer (ViT).** ViT (Dosovitskiy et al., 2020) has demonstrated exceptional performance on various vision tasks. In video processing, TimeSformer (Bertasius et al., 2021) investigates the factorization of spatial and temporal self-attention and proposes that divided attention where temporal and spatial attention are applied separately yields the best accuracy. ViViT (Arnab et al., 2021) explores factorized encoders, self-attention, and dot product mechanisms, concluding that a factorized encoder with spatial attention applied first performs better. On the other hand, TSViT (Tarasiou et al., 2023) finds that a factorized encoder prioritizing temporal attention achieves superior results. Latte (Ma et al., 2024) investigates factorized encoders and factorized self-attention mechanisms, incorporating both spatial-first and spatial-temporal parallel designs, within the context of latent diffusion transformers for video generation. Despite these advancements, most existing models focus primarily on video classification, with limited research on applying ViTs to spatio-temporal predictive learning. Moving beyond earlier methods that focus on factorizing self-attention, PredFormer explores the decomposition of spatial and temporal transformers at a deeper level by integrating self-attention with gated linear units and introducing innovative interleaved designs, allowing for a more robust capture of complex spatiotemporal dynamics.

## 3 Method

To systematically analyze the transformer structure of the network model in spatial-temporal predictive learning, we propose the PredFormer as a general model design, as shown in Fig 2. In the following sections, we introduce the pure transformer-based architecture in Sec 3.1. Next, we describe the Gated Transformer Block (GTB) in Sec 3.2. Finally, we present how to use GTB to build a PredFormer layer and architecture variants in Sec 3.3.

### 3.1 Pure Transformer Based Architecture

**Patch Embedding.** Follow the ViT design, PredFormer splits a sequence of frames $\mathcal{X}$ into a sequence of $N = \left\lfloor \frac{H}{p} \right\rfloor \left\lfloor \frac{W}{p} \right\rfloor$ equally sized, non-overlapping patches of size $p$, each of which is flattened into a 1D tokens.

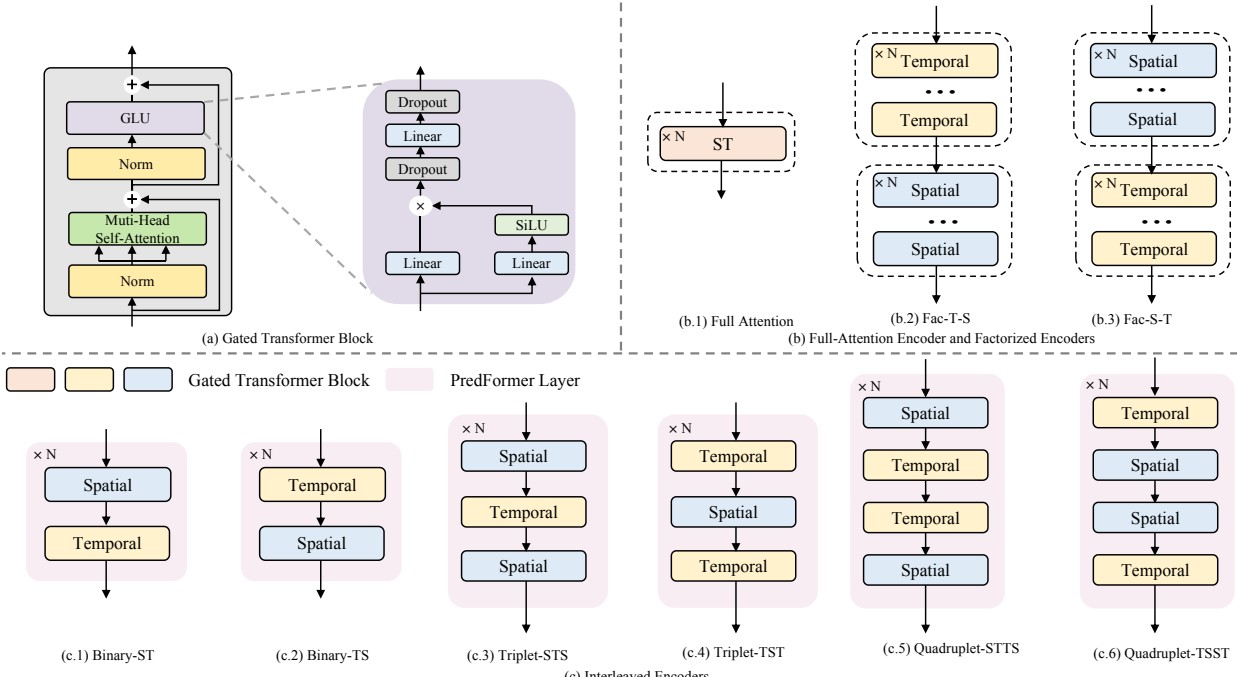

Figure 3: (a) Gated Transformer Block (b) Full Attention Encoder and Factorized Encoders (c) Interleaved Encoders with Binary, Triplet, and Quadrupled design

These tokens are then linearly projected into hidden dimensions $D$ and processed by a layer normalization (LN) layer, resulting in a tensor $\mathcal{X}' \in \mathbb{R}^{B \times T \times N \times D}$.

**Position Encoding.** Unlike typical ViT approach, which employs learnable position embeddings, we incorporate a spatiotemporal position encoding (PE) generated by sinusoidal functions with absolute coordinates for each patch.

**PredFormer Encoder.** The 1D tokens are then processed by a PredFormer Encoder for feature extraction. PredFormer Encoder is stacked by Gated Transformer Blocks in various manners.

**Patch Recovery.** Since our encoder is based on a pure gated transformer, without convolution or resolution reduction, global context is modeled at every layer. This allows it to be paired with a simple decoder, forming a powerful prediction model. After the encoder, a linear layer serves as the decoder, projecting the hidden dimensions back to recover the 1D tokens to 2D patches.

## 3.2 Gated Transformer Block

The Standard Transformer model (Vaswani et al., 2017) alternates between Multi-Head Attention (MSA) and Feed-Forward Networks (FFN). The attention mechanism for each head is defined as:

$$\text{Attention}(\mathbf{Q}, \mathbf{K}, \mathbf{V}) = \text{Softmax}\left(\frac{\mathbf{Q}\mathbf{K}^\top}{\sqrt{d_k}}\right)\mathbf{V}, \tag{1}$$

where in self-attention, the queries $\mathbf{Q}$, keys $\mathbf{K}$, and values $\mathbf{V}$ are linear projections of the input $\mathbf{X}$, represented as $\mathbf{Q} = \mathbf{X}\mathbf{W}_q$, $\mathbf{K} = \mathbf{X}\mathbf{W}_k$, and $\mathbf{V} = \mathbf{X}\mathbf{W}_v$, with $\mathbf{X}, \mathbf{Q}, \mathbf{K}, \mathbf{V} \in \mathbb{R}^{N \times d}$. The FFN then processes each position in the sequence by applying two linear transformations.

Gated Linear Units (GLUs) (Dauphin et al., 2017), often used in place of simple linear transformations, involve the element-wise product of two linear projections, with one projection passing through a sigmoid function. Various GLU variants control the flow of information by substituting the sigmoid with other non-linear functions. For instance, SwiGLU (Shazeer, 2020) replaces the sigmoid with the Swish activation

Table 1: Benchmark datasets used in our experiments. "Interval" denotes the temporal gap between two consecutive frames in the sequence (e.g., frame-level sampling, 30 minutes, or 1 hour).

| Dataset | Training size | Testing size | Channel | Height | Width | Input $T$ | Output $T'$ | Interval |
|---|---|---|---|---|---|---|---|---|
| Moving MNIST | 10,000 | 10,000 | 1 | 64 | 64 | 10 | 10 | - |
| Human3.6m | 73,404 | 8,582 | 3 | 256 | 256 | 4 | 4 | frame |
| WeatherBench-S | 52,559 | 17,495 | 1 | 32 | 64 | 12 | 12 | 30 min |
| TaxiBJ | 20,461 | 500 | 2 | 32 | 32 | 4 | 4 | 1 hour |

function (SiLU) (Hendrycks & Gimpel, 2016), as shown in Eq 2.

$$\text{Swish}_\beta(x) = x\sigma(\beta x)$$
$$\text{SwiGLU}(x, W, V, b, c, \beta) = \text{Swish}_\beta(xW + b) \otimes (xV + c) \tag{2}$$

SwiGLU has been demonstrated to outperform Multi-layer Perceptrons (MLPs) in various natural language processing tasks(Shazeer, 2020). Inspired by the SwiGLU's success in these tasks, our Gated Transformer Block (GTB), incorporates MSA followed by a SwiGLU-based FFN, as illustrated in Fig 3(a). GTB is defined as:

$$\mathbf{Y}^l = \text{MSA}(\text{LN}(\mathbf{Z}^l)) + \mathbf{Z}^l$$
$$\mathbf{Z}^{l+1} = \text{SwiGLU}(\text{LN}(\mathbf{Y}^l)) + \mathbf{Y}^l \tag{3}$$

### 3.3 Variants of PredFormer

Modeling spatiotemporal dependencies in video prediction is challenging, as the balance between spatial and temporal information differs significantly across tasks and datasets. Developing flexible and adaptive models that can accommodate varying dependencies and scales is thus critical. To address these, we explore both full-attention encoders and factorized encoders with spatial-first (Fac-S-T) and temporal-first (Fac-T-S) configurations, as shown in Fig 3(b). In addition, we introduce six interleaved models based on PredFormer layer, enabling dynamic interaction across multiple scales.

A PredFormer layer is a module capable of simultaneously processing spatial and temporal information. Building on this design principle, we propose three interleaved spatiotemporal paradigms, Binary, Triplet, and Quadruplet, which sequentially model the spatial and temporal views. Ultimately, they yield **six** distinct architectural configurations. A detailed illustration of these nine variants is provided in Fig 3.

For full attention layers, given input $\mathcal{X} \in \mathbb{R}^{B \times T \times N \times D}$, attention is computed over the sequence of length $T \times N$. As illustrated in Fig 3 (b.1), we merge and flatten the spatial and temporal tokens to compute attention through several stacked $\text{GTB}_{st}$.

For Binary layers, each GTB block processes temporal or spatial sequence independently, which we denote as a binary-TS or binary-ST layer. The input is first reshaped, and processed through $\text{GTB}_t^1$, where attention is applied over the temporal sequence. The tensor is then reshaped back to restore the temporal order. Subsequently, spatial attention is applied using another $\text{GTB}_s^2$, where the tensor is flattened along the temporal dimension and processed.

For Triplet and Quadruplet layers, additional blocks are stacked on top of the Binary structure. The Quadruplet layer combines two Binary layers in different orders.

## 4 Experiments

We present extensive evaluations of PredFormer and state-of-the-art methods. We conduct experiments across synthetic and real-world scenarios, including long-term prediction(moving object trajectory prediction and weather forecasting), and short-term prediction(traffic flow prediction and human motion prediction). The statistics of the data set are presented in the tab 1. These datasets have different spatial resolutions, temporal frames, and intervals, which determine their different spatiotemporal dependencies.

Table 2: Quantitative comparison on **Moving MNIST**. Each model observes 10 frames and predicts the subsequent 10 frames. We highlight the best experimental results in bold red and the second-best in blue.

| Method | Paras(M) | Flops(G) | FPS | Memory (MB) | MSE ↓ | MAE ↓ | SSIM ↑ |
|---|---|---|---|---|---|---|---|
| ConvLSTM | 15.0 | 56.8 | 113 | 68.8 | 103.3 | 182.9 | 0.707 |
| PredRNN | 23.8 | 116.0 | 54 | 107.6 | 56.8 | 126.1 | 0.867 |
| PredRNN++ | 38.6 | 171.7 | 38 | 164.1 | 46.5 | 106.8 | 0.898 |
| MIM | 38.0 | 179.2 | 37 | 160.6 | 44.2 | 101.1 | 0.910 |
| E3D-LSTM | 51.0 | 298.9 | 18 | 270.1 | 41.3 | 86.4 | 0.910 |
| PhyDNet | 3.1 | 15.3 | 182 | 149.5 | 24.4 | 70.3 | 0.947 |
| MAU | 4.5 | 17.8 | 201 | 35.4 | 27.6 | 86.5 | 0.937 |
| PredRNNv2 | 24.6 | 708.0 | 24 | 109.0 | 48.4 | 129.8 | 0.891 |
| SwinLSTM | 20.2 | 69.9 | 62 | 96.4 | 17.7 | - | 0.962 |
| SimVP | 58.0 | 19.4 | 209 | 284.7 | 23.8 | 68.9 | 0.948 |
| TAU | 44.7 | 16.0 | 283 | 322.9 | 19.8 | 60.3 | 0.957 |
| OpenSTL_ViT | 46.1 | 16.9 | 290 | 331.8 | 19.0 | 60.8 | 0.955 |
| OpenSTL_Swin | 46.1 | 16.9 | 290 | 331.9 | 18.3 | 59.0 | 0.960 |
| **PredFormer** | | | | | | | |
| Full Attention | 25.3 | 21.2 | 254 | 135.9 | 17.3 | 56.0 | 0.962 |
| Fac-S-T | 25.3 | 16.5 | 368 | 117.4 | 20.6 | 63.5 | 0.955 |
| Fac-T-S | 25.3 | 16.5 | 370 | 117.4 | 16.9 | 55.8 | 0.963 |
| Binary-TS | 25.3 | 16.5 | 301 | 117.4 | 12.8 | 46.1 | 0.972 |
| Binary-ST | 25.3 | 16.5 | 316 | 117.4 | 13.4 | 47.1 | 0.971 |
| Triplet-TST | 25.3 | 16.4 | 312 | 118.0 | 13.4 | 47.2 | 0.971 |
| Triplet-STS | 25.3 | 16.5 | 321 | 118.0 | 13.1 | 46.7 | 0.972 |
| Quadruplet-TSST | 25.3 | 16.5 | 302 | 118.0 | 12.4 | 44.6 | 0.973 |
| Quadruplet-STTS | 25.3 | 16.4 | 322 | 118.0 | 12.4 | 44.9 | 0.973 |

**Implementation Details** Our method is implemented in PyTorch. The experiments were conducted on a single 24GB NVIDIA RTX 3090. PredFormer is optimized using the AdamW (Loshchilov & Hutter, 2019) optimizer with an L2 loss, a weight decay of 1e-2, and a learning rate selected from {5e-4, 1e-3} for best performance. OneCycle scheduler is used for Moving MNIST and TaxiBJ, while the Cosine scheduler is applied for Human3.6m and WeatherBench. Dropout (Hinton, 2012) and stochastic depth (Huang et al., 2016) regularization prevent overfitting. Further hyperparameter details are provided in the Appendix. For different PredFormer variants, we maintain a constant number of GTB blocks to ensure comparable parameters. In cases where the Triplet model cannot be evenly divided, we use the number of GTB blocks closest to the others.

**Evaluation Metrics** We assess model performance using a suite of metrics. (1) **Pixel-wise error** is measured using Mean Squared Error (MSE), Mean Absolute Error (MAE), and Root Mean Squared Error (RMSE). (2) **Predicted frame quality** is evaluated using the structural similarity index measure (SSIM) metric (Wang et al., 2004). Lower MSE, MAE, and RMSE values, combined with higher SSIM, signify better predictions. (3) **Computational efficiency** is assessed by the number of parameters, floating-point operations (FLOPs), and inference speed in frames per second (FPS) on a NVIDIA A6000 GPU. This evaluation framework comprehensively evaluates accuracy and efficiency.

## 4.1 Synthetic Moving Object Prediction

**Moving MNIST.** The moving MNIST dataset (Srivastava et al., 2015) serves as a benchmark synthetic dataset for evaluating sequence reconstruction models. We follow (Srivastava et al., 2015) to generate Moving MNIST sequences with 20 frames, using the initial 10 frames for input and the subsequent 10 frames as the prediction target. We adopt 10000 sequences for training, and for fair comparisons, we use the pre-generated 10000 sequences (Gao et al., 2022a) for validation.

On the Moving MNIST dataset, following prior work (Gao et al., 2022a; Tan et al., 2023a), we train our models for 2000 epochs and report our results in Tab 2. We train OpenSTL methods with ViT and Swin Transformer as temporal translators for 2000 epochs as recurrent-free baselines. We cite other results from each original paper for a fair comparison.

Compared to SimVP, PredFormer achieves substantial performance gains while maintaining a lightweight structure. Specifically, it reduces MSE by 48% (from 23.8 to 12.4), significantly improving prediction accu-

Table 3: Quantitative comparison on **Human3.6m**. Each model observes 4 frames and predicts the subsequent 4 frames.

| Method | Paras(M) | Flops(G) | FPS | Memory(MB) | MSE ↓ | MAE ↓ | SSIM ↑ | PSNR ↑ | LPIPS ↓ |
|---|---|---|---|---|---|---|---|---|---|
| ConvLSTM | 15.5 | 347.0 | 52 | 142.7 | 125.5 | 1566.7 | 0.9813 | 33.40 | 0.03557 |
| PredNet | 12.5 | 13.7 | 176 | 120.8 | 261.9 | 1625.3 | 0.9786 | 31.76 | 0.03264 |
| PredRNN | 24.6 | 704.0 | 25 | 327.2 | 113.2 | 1458.3 | 0.9831 | 33.94 | 0.03245 |
| PredRNN++ | 39.3 | 1033.0 | 18 | 402.3 | 110.0 | 1452.2 | 0.9832 | 34.02 | 0.03196 |
| MIM | 47.6 | 1051.0 | 17 | 434.8 | 112.1 | 1467.1 | 0.9829 | 33.97 | 0.03338 |
| E3D-LSTM | 60.9 | 542.0 | 7 | 548.9 | 143.3 | 1442.5 | 0.9803 | 32.52 | 0.04133 |
| PhyDNet | 4.2 | 19.1 | 57 | 67.9 | 125.7 | 1614.7 | 0.9804 | 33.05 | 0.03709 |
| MAU | 20.2 | 105.0 | 6 | 371.2 | 127.3 | 1577.0 | 0.9812 | 33.33 | 0.03561 |
| PredRNNv2 | 24.6 | 708.0 | 24 | 350.5 | 114.9 | 1484.7 | 0.9827 | 33.84 | 0.03334 |
| SimVP | 41.2 | 197.0 | 26 | 556.4 | 115.8 | 1511.5 | 0.9822 | 33.73 | 0.03467 |
| TAU | 37.6 | 182.0 | 26 | 551.8 | 113.3 | 1390.7 | 0.9839 | 34.03 | 0.02783 |
| OpenSTL_ViT | 11.0 | 142.2 | 35 | 1170.0 | 136.3 | 1603.5 | 0.9796 | 33.10 | 0.03729 |
| OpenSTL_Swin | 38.8 | 188.0 | 28 | 562.3 | 133.2 | 1599.7 | 0.9799 | 33.16 | 0.03766 |
| **PredFormer** | | | | | | | | | |
| Full Attention | 12.7 | 155.0 | 16 | 1120.7 | 113.9 | 1412.4 | 0.9833 | 33.98 | 0.03279 |
| Fac-S-T | 12.7 | 65.0 | 76 | 352.7 | 153.4 | 1630.7 | 0.9784 | 32.30 | 0.04676 |
| Fac-T-S | 12.7 | 65.0 | 75 | 356.7 | 118.4 | 1504.7 | 0.9820 | 33.67 | 0.03284 |
| Binary-TS | 12.7 | 65.0 | 75 | 352.7 | 111.2 | 1380.4 | 0.9838 | 34.13 | 0.03008 |
| Binary-ST | 12.7 | 65.0 | 78 | 348.7 | 112.7 | 1386.3 | 0.9836 | 34.07 | 0.03017 |
| Triplet-TST* | 12.7 | 60.8 | 88 | 352.7 | 112.4 | 1406.2 | 0.9834 | 34.05 | 0.02748 |
| Triplet-STS* | 12.7 | 69.3 | 64 | 356.7 | 111.8 | 1410.3 | 0.9834 | 34.07 | 0.02933 |
| Quadruplet-TSST | 12.7 | 65.0 | 72 | 352.7 | 110.9 | 1380.3 | 0.9839 | 34.14 | 0.03069 |
| Quadruplet-STTS | 12.7 | 65.0 | 74 | 356.7 | 113.4 | 1405.7 | 0.9835 | 34.04 | 0.02918 |

For * models, we add a skip connection for each PredFormer Layer for stable training.

racy. Meanwhile, PredFormer requires far fewer parameters (25.3M vs. 58.0M in SimVP) and operates with lower FLOPs (16.5G vs. 19.4G), showcasing its superior efficiency.

Notably, even when SimVP incorporates ViT and Swin Transformer as the temporal translator, its performance remains far below that of PredFormer. This is because, while SimVP benefits from the inductive bias of using CNNs as the encoder and decoder, this design inherently limits the model's performance ceiling. In contrast, PredFormer effectively models global spatiotemporal dependencies, allowing it to surpass these constraints and achieve superior predictive accuracy.

Compared to SwinLSTM, PredFormer achieves higher accuracy. Although SwinLSTM outperforms SimVP in terms of MSE (17.7 vs. 23.8), its reliance on an RNN-based structure results in significantly higher computational cost. SwinLSTM exhibits high FLOPs (69.9G) and lower FPS, making it less efficient for large-scale deployment. This highlights the limitations of recurrent structures in video prediction, whereas PredFormer, with its recurrence-free framework, achieves both higher accuracy and superior efficiency.

Among the PredFormer variants, Quadruplet-TSST achieves the best MSE of 12.4, followed closely by Quadruplet-STTS. These results highlight PredFormer's ability to fully leverage global information fully, further validating its effectiveness in video prediction.

## 4.2 Real-world Human Motion Prediction

**Human3.6m.** The Human3.6M dataset (Ionescu et al., 2014) comprises 3.6 million unique human poses with their corresponding images, serving as a benchmark for motion prediction tasks. Human motion prediction is particularly challenging due to high resolution input and complex human movement dynamics. Following OpenSTL (Tan et al., 2023b), we downsample the dataset from $1000\times1000\times3$ to $256\times256\times3$. We use four observations to predict the next four frames.

PredFormer achieves SOTA on Human3.6M. Compared to SimVP, PredFormer Quadruplet-TSST reduces MSE from 115.8 to 110.9, significantly improving human motion prediction. At the same time, PredFormer requires only 12.7M parameters, much fewer than SimVP's 41.2M. Furthermore, its computational cost is only 65G FLOPs, less than one third of SimVP's 197G, while maintaining a higher inference speed.

PredFormer also substantially reduces the computational cost compared to PredRNN++. While PredRNN++ requires 1033G FLOPs, PredFormer Quadruplet-TSST achieves comparative MSE and superior MAE and SSIM using only one-tenth of the computation.

Table 4: Quantitative comparison on **TaxiBJ**. Each model observes 4 frames and predicts the subsequent 4 frames.

| Method | Paras(M) | Flops(G) | FPS | Memory(MB) | MSE ↓ | MAE ↓ | SSIM ↑ |
|---|---|---|---|---|---|---|---|
| ConvLSTM | 15.0 | 20.7 | 815 | 67.0 | 0.485 | 17.7 | 0.978 |
| PredRNN | 23.7 | 42.4 | 416 | 105.5 | 0.464 | 16.9 | 0.977 |
| PredRNN++ | 38.4 | 63.0 | 301 | 162.8 | 0.448 | 16.9 | 0.971 |
| MIM | 37.9 | 64.1 | 275 | 158.4 | 0.429 | 16.6 | 0.971 |
| E3D-LSTM | 51.0 | 98.2 | 60 | 240.5 | 0.432 | 16.9 | 0.979 |
| PhyDNet | 3.1 | 5.6 | 982 | 149.3 | 0.362 | 15.5 | 0.983 |
| PredRNNv2 | 23.7 | 42.6 | 378 | 106.8 | 0.383 | 15.5 | 0.983 |
| SwinLSTM | 2.9 | 1.3 | 1425 | 22.0 | 0.303 | 15.0 | 0.984 |
| SimVP | 13.8 | 3.6 | 533 | 183.9 | 0.414 | 16.2 | 0.982 |
| TAU | 9.6 | 2.5 | 1268 | 175.0 | 0.344 | 15.6 | 0.983 |
| OpenSTL_ViT | 9.7 | 2.8 | 1301 | 174.8 | 0.317 | 15.2 | 0.984 |
| OpenSTL_Swin | 9.7 | 2.6 | 1506 | 274.3 | 0.313 | 15.1 | 0.985 |
| **PredFormer** | | | | | | | |
| Full Attention | 8.4 | 2.4 | 2438 | 42.3 | 0.316 | 14.6 | 0.985 |
| Fac-S-T | 8.4 | 2.2 | 3262 | 42.4 | 0.320 | 15.2 | 0.984 |
| Fac-T-S | 8.4 | 2.2 | 3224 | 42.4 | 0.283 | 14.4 | 0.985 |
| Binary-TS | 8.4 | 2.2 | 3192 | 42.4 | 0.286 | 14.6 | 0.985 |
| Binary-ST | 8.4 | 2.2 | 3172 | 42.4 | **0.277** | **14.3** | **0.986** |
| Triplet-TST | 6.3 | 1.6 | 4348 | 34.4 | 0.293 | 14.7 | 0.985 |
| Triplet-STS | 6.3 | 1.6 | 4249 | 34.4 | **0.277** | **14.3** | **0.986** |
| Quadruplet-TSST | 8.4 | 2.2 | 3230 | 42.4 | 0.284 | 14.4 | **0.986** |
| Quadruplet-STTS | 8.4 | 2.2 | 3259 | 42.4 | 0.293 | 14.6 | 0.985 |

Compared to OpenSTL-ViT and OpenSTL-Swin Transformer, which rely on ViT-based architectures but struggle with prediction accuracy, PredFormer utilizes the Transformer structures more effectively for video prediction. OpenSTL-ViT and OpenSTL-Swin both perform worse than SimVP, indicating that simply applying Transformers does not guarantee strong results. In contrast, PredFormer outperforms them while maintaining an efficient design, demonstrating its capability in spatiotemporal modeling.

### 4.3 Traffic Flow Prediction

**TaxiBJ.** TaxiBJ (Zhang et al., 2017a) includes GPS data from taxis and meteorological data in Beijing. Each data frame is visualized as a $32 \times 32 \times 2$ heatmap, where the third dimension encapsulates the inflow and outflow of traffic within a designated area. Following previous work (Zhang et al., 2017a), we allocate the final four weeks' data for testing, utilizing the preceding data for training. Our prediction model uses four sequential observations to forecast the subsequent four frames.

PredFormer achieves SOTA on TaxiBJ. Compared to SimVP, PredFormer significantly improves the accuracy of the prediction, reducing the MSE from 0.414 to 0.277 (33%) while using fewer parameters (8.4M vs 13.8M) and a lower computational cost (2.2G FLOPs vs. 3.6G FLOPs). Despite this efficiency, PredFormer also dramatically increases inference speed, with FPS rising from 533 in SimVP to 4249.

Furthermore, OpenSTL-ViT and OpenSTL-Swin, which adopt ViT-based architectures as temporal translators, achieve MSEs of 0.317 and 0.313, both worse than PredFormer's best results. This suggests that using CNNs for the encoder and decoder provides a strong inductive bias but inherently limits the model's performance. Among the variants of PredFormer, Binary-ST and Triplet-STS achieve the best MSE of 0.277.

### 4.4 Weather Forecasting

**WeatherBench.** Climate prediction is a critical challenge in spatiotemporal predictive learning. The WeatherBench (Rasp et al., 2020) dataset provides a comprehensive global weather forecasting resource, covering various climatic factors. In our experiments, we utilize *WeatherBench-S*, a single-variable setup where each climatic factor is trained independently. We focus on temperature prediction at a $5.625°$ resolution ($32 \times 64$ grid points). The model is trained on data spanning 2010-2015, validated on data from 2016, and tested on data from 2017-2018, all with a one-hour temporal interval. We input the first 12 frames and predict the subsequent 12 frames in this setting.

Table 5: Quantitative comparison on **WeatherBench(T2m)**. Each model observes 12 frames and predicts the subsequent 12 frames.

| Method | Paras(M) | Flops(G) | FPS | Memory(MB) | MSE ↓ | MAE ↓ | SSIM ↑ |
|---|---|---|---|---|---|---|---|
| ConvLSTM | 14.9 | 136.0 | 46 | 69.8 | 1.521 | 0.7949 | 1.233 |
| PredRNN | 23.6 | 278.0 | 22 | 108.2 | 1.331 | 0.7246 | 1.154 |
| PredRNN++ | 38.3 | 413.0 | 15 | 165.5 | 1.634 | 0.7883 | 1.278 |
| MIM | 37.8 | 109.0 | 126 | 275.9 | 1.784 | 0.8716 | 1.336 |
| PhyDNet | 3.1 | 36.8 | 177 | 150.6 | 285.9 | 8.7370 | 16.91 |
| MAU | 5.5 | 39.6 | 237 | 56.2 | 1.251 | 0.7036 | 1.119 |
| PredRNNv2 | 23.6 | 279.0 | 22 | 110.8 | 1.545 | 0.7986 | 1.243 |
| SimVP | 14.8 | 8.0 | 196 | 194.7 | 1.238 | 0.7037 | 1.113 |
| TAU | 12.2 | 6.7 | 229 | 195.6 | 1.162 | 0.6707 | 1.078 |
| OpenSTL_ViT | 12.4 | 8.0 | 432 | 194.9 | 1.146 | 0.6712 | 1.070 |
| OpenSTL_Swin | 12.4 | 6.9 | 581 | 195.1 | 1.143 | 0.6735 | 1.069 |
| **PredFormer** | | | | | | | |
| Full Attention | 5.3 | 17.8 | 177 | 185.4 | 1.126 | 0.6540 | 1.061 |
| Fac-S-T | 5.3 | 8.5 | 888 | 53.9 | 1.783 | 0.8688 | 1.335 |
| Fac-T-S | 5.3 | 8.5 | 860 | 54.9 | **1.100** | **0.6469** | **1.049** |
| Binary-TS | 5.3 | 8.6 | 837 | 53.9 | 1.115 | 0.6508 | 1.056 |
| Binary-ST | 5.3 | 8.6 | 847 | 51.9 | 1.140 | 0.6571 | 1.068 |
| Triplet-TST | 4.0 | 6.3 | 1064 | 48.9 | 1.108 | 0.6492 | 1.053 |
| Triplet-STS | 4.0 | 6.5 | 1001 | 49.9 | 1.149 | 0.6658 | 1.072 |
| Quadruplet-TSST | 5.3 | 8.6 | 802 | 53.9 | 1.116 | 0.6510 | 1.057 |
| Quadruplet-STTS | 5.3 | 8.6 | 858 | 54.9 | 1.118 | 0.6507 | 1.057 |

Table 6: Ablation on PredFormer Layer Number on Moving MNIST.

| TSST Layer | Paras(M) | Flops(G) | FPS | MSE ↓ | MAE ↓ | SSIM ↑ |
|---|---|---|---|---|---|---|
| 2 | 8.5 | 5.5 | 887 | 20.1 | 65.2 | 0.955 |
| 3 | 12.7 | 8.3 | 621 | 16.2 | 55.1 | 0.965 |
| 4 | 16.9 | 11.0 | 457 | 13.5 | 47.9 | 0.970 |
| 5 | 21.1 | 13.7 | 376 | 12.7 | 45.5 | 0.972 |
| 6 | 25.3 | 16.5 | 302 | 12.4 | 44.6 | 0.973 |
| 7 | 29.5 | 19.2 | 277 | 12.3 | 44.2 | 0.973 |
| 8 | 33.7 | 22.0 | 240 | **11.7** | **41.8** | **0.975** |

On WeatherBench (T2m), RNN-based models like ConvLSTM and PredRNN have high computational costs but poor performance. PredRNN and PredRNNv2 reach 278.0G and 279.0G FLOPs, yet their MSE remains at 1.331 and 1.545, respectively, highlighting the inefficiency of RNN structures for this task.

SimVP reduces parameter count to half of PredRNN, lowers FLOPs to 8.0G, and achieves an improved MSE of 1.238, making it more efficient than RNN models. PredFormer further improves performance, using only half the parameters of SimVP while maintaining similar or lower FLOPs, achieving the best MSE of 1.100. The Fac-T-S and Triplet-TST variants deliver the top results. It also demonstrates a significant FPS advantage, with Binary-TS and Triplet-TST achieving 837 and 1064 FPS, respectively, compared to SimVP's 196, highlighting the model's superior efficiency in both computation and prediction speed.

### 4.5   Ablation Study and Discussion

**PredFormer Layer Number.** We conduct an ablation study on the number of TSST layers in PredFormer to evaluate its scalability and potential for performance improvement, as shown in Tab 6. The results show that as the number of layers increases, PredFormer continues to achieve better results, surpassing the 6-layer TSST configuration reported in Tab 2. With 2 TSST layers, PredFormer already outperforms SimVP, achieving a lower MSE of 20.1 while maintaining high efficiency. When increasing to 3 layers, PredFormer surpasses TAU, OpenSTL-ViT, and OpenSTL-Swin, achieving a lower MSE of 16.2 while requiring only half the FLOPs of these models and delivering twice their FPS. With 8 layers, PredFormer achieves an MSE of 11.7, which represents a 51% MSE reduction compared to SimVP. This substantial improvement demonstrates the scalability of PredFormer.

We conduct ablation studies on PredFormer model design and summarize the results in Tab 7 and Tab 8. We choose the best Triplet-STS model on TaxiBJ, and the best Fac-T-S model on WeatherBench as baselines.

Table 7: Ablation on Gate Linear Unit and Position Encoding.

| Model | WeatherBench (T2m) | | | TaxiBJ | |
|---|---|---|---|---|---|
| | MSE ↓ | MAE ↓ | RMSE ↓ | MSE ↓ | MAE ↓ |
| PredFormer | **1.100** | **0.6489** | **1.049** | **0.277** | **14.3** |
| SwiGLU → MLP | 1.171 | 0.6707 | 1.082 | 0.306 | 15.1 |
| PE: Abs → Learnable | 1.164 | 0.6771 | 1.079 | 0.288 | 14.6 |

Table 8: Ablation on Dropout and Stochastic Depth.

| Model | WeatherBench (T2m) | | | TaxiBJ | |
|---|---|---|---|---|---|
| | MSE ↓ | MAE ↓ | RMSE ↓ | MSE ↓ | MAE ↓ |
| + DP + Uni SD | **1.100** | **0.6489** | **1.049** | **0.277** | **14.3** |
| W/o Reg | 1.244 | 0.7057 | 1.115 | 0.319 | 15.1 |
| + DP | 1.210 | 0.6887 | 1.100 | 0.283 | 14.5 |
| + Uni SD | 1.156 | 0.6573 | 1.075 | 0.288 | 14.6 |
| + DP + Linear SD | 1.138 | 0.6533 | 1.067 | 0.299 | 14.8 |

**Gate Linear Unit.** Replacing SwiGLU with a standard MLP results in a notable performance degradation. On TaxiBJ, the MSE rises from 0.277 to 0.306, and on WeatherBench from 1.100 to 1.171. This consistent performance degradation highlights the critical role of the gating mechanism in modeling complex spatiotemporal dynamics.

**Position Encoding.** Additionally, the performance deteriorates when we replace the absolute positional encoding in our model with the learnable spatiotemporal encoding commonly used in ViT. On Moving TaxiBJ, the MSE rises from 0.277 to 0.288, and on WeatherBench from 1.100 to 1.164. These ablation experiments consistently reveal similar trends across all three datasets, emphasizing the robustness of our Position Encoding designs.

**Model Regularization.** Pure transformer architectures like ViT generally require large datasets for effective training, and overfitting can become challenging when applied to smaller datasets. In our experiments, overfitting is noticeable on WeatherBench and TaxiBJ. We experiment with different regularization techniques in Tab 8 and find that both dropout(DP) and stochastic depth (SD) individually improve performance compared to no regularization. However, the combination of the two provides the best results. Unlike conventional ViT practices, which use a linearly scaled drop path rate across different depths, a uniform drop path rate performs significantly better for our tasks.

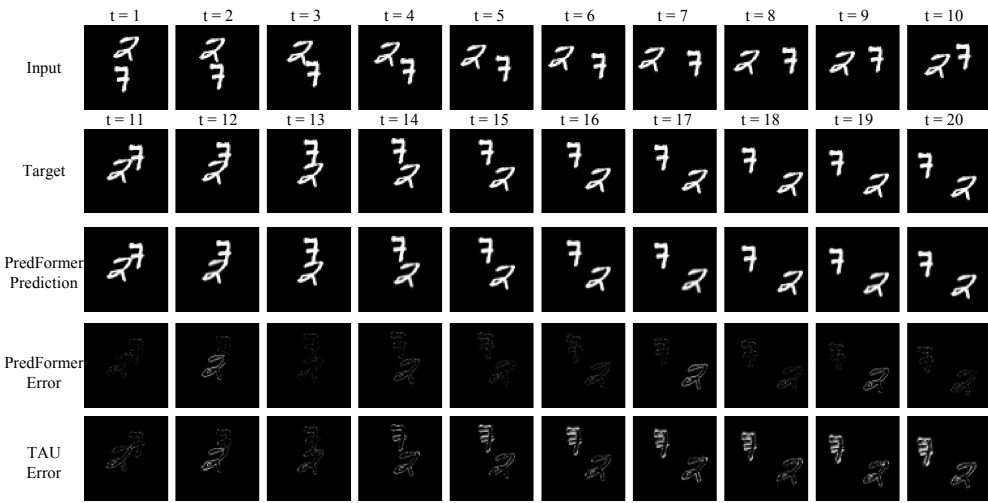

Figure 4: Visualizations on Moving MNIST. Error = |Prediction − Target|. We amplify the error for better comparison.

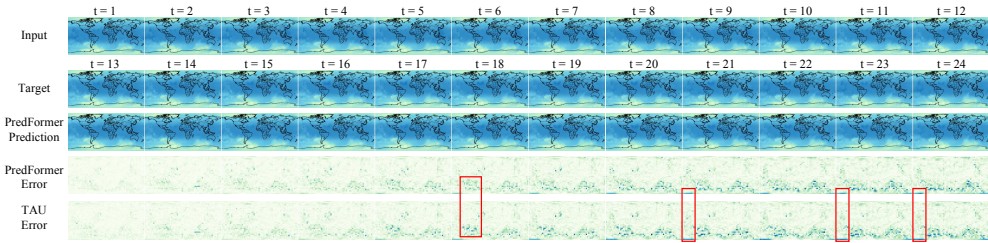

Figure 5: Visualizations on WeatherBench for global temperature forecasting.

**Visualization.** Fig 4, 5 and 6 provide a visual comparison of PredFormer's prediction results and prediction errors with Ground Truth. For Moving MNIST, our model accurately captures digit trajectories, with significantly lower accumulated error compared to TAU. On TaxiBJ, PredFormer effectively reconstructs the intricate spatial structures of traffic patterns, reducing high-frequency noise present in TAU's predictions. On WeatherBench, PredFormer achieves sharper and more precise temperature forecasts, with error heatmaps showing lower deviations in critical regions. Lastly, for Human3.6m, PredFormer consistently preserves fine-grained motion details, demonstrating superior temporal coherence in video prediction. Additional visualizations are provided in the supplementary material.

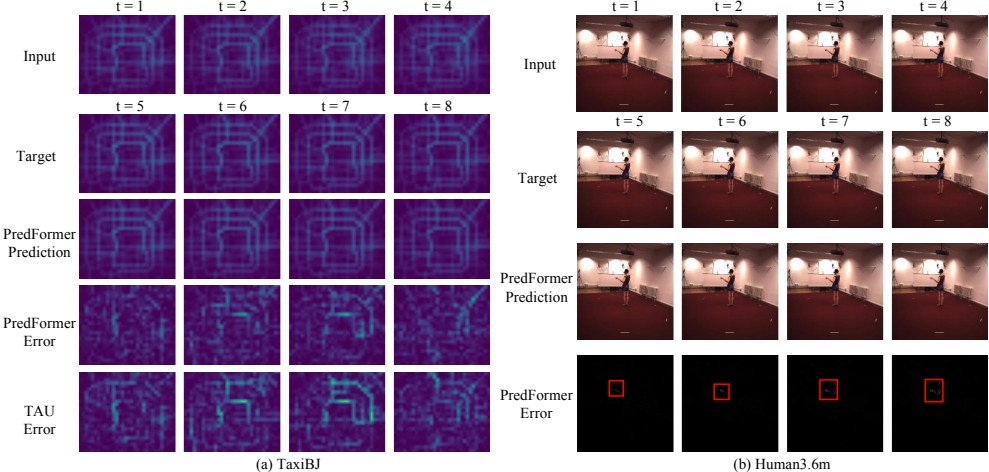

Figure 6: Visualizations on TaxiBJ and Human3.6m.

**Discussion for PredFormer Variants.** Despite our in-depth analysis of the spatiotemporal decomposition, the optimal model is not definite due to the different spatiotemporal dependent properties of the datasets. We recommend starting with the Quadruplet-TSST model for diverse video prediction tasks, which consistently performs well across datasets and configurations. Use M Quadruplet-TSST layers and experiment with models having a total of 4M GTBs to identify the optimal configuration. Then, explore Triplet-TST and Triplet-STS with M layers to find spatial and temporal dependencies. Unlike the SimVP framework, which adjusts hidden dimensions and block numbers separately for spatial encoder-decoder and temporal translator, PredFormer uses fixed hyperparameters for spatial and temporal GTBs, leveraging the scalability of the Transformer architecture. By simply adjusting the number of PredFormer layers, optimal results can be achieved with minimal tuning. We provide further theoretical analysis in the Appendix.

## 5 Conclusion

In this paper, we introduce PredFormer, a pure Transformer-based framework for video prediction, and systematically study how different spatio-temporal factorization patterns affect efficiency and accuracy. Across four standard benchmarks, PredFormer achieves strong performance while maintaining favorable

FLOPs, FPS, and memory usage compared to prior CNN- and RNN-based models, providing a competitive recurrent-free and convolution-free alternative. On the four benchmarks we study, our experiments suggest several dataset-dependent tendencies: interleaved spatio-temporal architectures generally offer the best balance between cost and accuracy; temporal-first factorizations often work particularly well for long-horizon forecasting, while other interleaved patterns can be preferable when the spatial structure is richer or the horizon is shorter; combining dropout with uniform stochastic depth is especially effective on overfitting-prone datasets; and absolute positional encoding is consistently more robust than learnable alternatives.

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

# A  Appendix

## A.1  Problem Definition

Video prediction is to learn spatial and temporal patterns by predicting future frames based on past observations. Given a sequence of frames $\mathcal{X}^{t:T} = \{\boldsymbol{x}^i\}_{t-T+1}^t$, which encapsulates the last $T$ frames leading up to time $t$, the goal is to forecast the following $T'$ frames $\mathcal{Y}^{t+1:T'} = \{\boldsymbol{x}^i\}_{t+1}^{t+1+T'}$ starting from time $t+1$. The input and the output sequence are represented as tensors $\mathcal{X}^{t:T} \in \mathbb{R}^{T \times C \times H \times W}$ and $\mathcal{Y}^{t+1:T'} \in \mathbb{R}^{T' \times C \times H \times W}$, where $C$, $H$, and $W$ denote channel, height, and width of frames, respectively. The $T$ and $T'$ are the input and output frame numbers. For brevity, we use $\mathcal{X}$ and $\mathcal{Y}$ to denote $\mathcal{X}^{t:T}$ and $\mathcal{Y}^{t+1:T'}$ in the following sections.

Generally, we adopt a deep model equipped with learnable parameters $\mathcal{F}_\Theta$ for future frame prediction. The optimal set of parameters $\Theta^*$ is obtained by solving the optimization problem:

$$\Theta^* = \arg\min_\Theta \mathcal{L}(\mathcal{F}_\Theta(\mathcal{X}), \mathcal{Y}) \tag{4}$$

where $\mathcal{L}$ is the loss function measuring the difference between the prediction and the ground truth.

## A.2  Data Transform

We provide a detailed description of the data transformation with PredFormer Binary-ST Layer in Eq 5. The data transformations for other variants follow a similar process.

$$
\begin{aligned}
[B, T, N, D] &\to [B * T, N, D], x_s = \text{GTB}_s^1(x_s.\text{flatten}(0, 1)) \\
[B * T, N, D] &\to [B, N, T, D], x_s = x_s.\text{reshape}(B, T, N, D).\text{T}(1, 2) \\
[B, N, T, D] &\to [B * N, T, D], x_{st} = \text{GTB}_t^2(x_s.\text{flatten}(0, 1)) \\
[B * N, T, D] &\to [B, T, N, D], x_{st} = x_{st}.\text{reshape}(B, N, T, D).\text{T}(1, 2)
\end{aligned}
\tag{5}
$$

## A.3  Theoretical Complexity Analysis

**Setup and notation.**  We provide a formal complexity analysis of the nine PredFormer variants in terms of the temporal length $T$, the number of spatial patches $N$, and the hidden dimension $D$. For a fair comparison, all variants are instantiated under the same overall attention budget: we fix the depth of the network (the number of spatio-temporal blocks) and only change how the self-attention operations inside each block are allocated along the temporal and spatial dimensions. In other words, the total number of self-attention blocks is kept comparable across variants, and the differences in complexity come from the factorization pattern, rather than from simply increasing the number of layers.

**Temporal vs. spatial self-attention.**  Let a temporal self-attention block operate on $N$ independent sequences of length $T$ (one for each spatial patch), and a spatial self-attention block operate on $T$ independent sequences of length $N$ (one for each time step). Ignoring constant factors and linear projections, the dominant terms are:

- **Temporal self-attention** over sequences of length $T$:

$$\text{Compute} \sim \mathcal{O}(T^2 N D), \qquad \text{Memory} \sim \mathcal{O}(T^2 N).$$

- **Spatial self-attention** over $N$ patches:

$$\text{Compute} \sim \mathcal{O}(T N^2 D), \qquad \text{Memory} \sim \mathcal{O}(T N^2).$$

- **Full spatio-temporal self-attention** over all $TN$ tokens:

$$\text{Compute} \sim \mathcal{O}((TN)^2 D), \qquad \text{Memory} \sim \mathcal{O}((TN)^2).$$

Table 9: Theoretical per-unit compute and memory complexity of the nine PredFormer variants and representative CNN/RNN baselines. For PredFormer, $T$ is the temporal length, $N$ is the number of spatial *patch* tokens (i.e., $N = HW/P^2$ for patch size $P$), and $D$ is the hidden dimension. For convolutional and recurrent baselines, we instead use the spatial grid size $H \cdot W$ (pixels or grid cells) to denote the number of spatial locations. $c_t$ and $c_s$ denote the number of temporal and spatial self-attention blocks in one spatio-temporal unit, and $c_j$ denotes the number of joint spatio-temporal attention blocks. We report only the leading terms and omit constant factors and lower-order terms.

| Variant / Baseline | $(c_t, c_s, c_j)$ | Compute Complexity | Memory Complexity |
|---|---|---|---|
| **PredFormer variants (patch tokens $N$)** | | | |
| Full Attention | $(0,0,1)$ | $\mathcal{O}\big((TN)^2 D\big)$ | $\mathcal{O}\big((TN)^2\big)$ |
| Fac-S-T | $(1,1,0)$ | $\mathcal{O}\big(TN^2 D + T^2 N D\big)$ | $\mathcal{O}\big(TN^2 + T^2 N\big)$ |
| Fac-T-S | $(1,1,0)$ | $\mathcal{O}\big(TN^2 D + T^2 N D\big)$ | $\mathcal{O}\big(TN^2 + T^2 N\big)$ |
| Binary-TS | $(1,1,0)$ | $\mathcal{O}\big(TN^2 D + T^2 N D\big)$ | $\mathcal{O}\big(TN^2 + T^2 N\big)$ |
| Binary-ST | $(1,1,0)$ | $\mathcal{O}\big(TN^2 D + T^2 N D\big)$ | $\mathcal{O}\big(TN^2 + T^2 N\big)$ |
| Triplet-TST | $(2,1,0)$ | $\mathcal{O}\big((2T^2 N + TN^2)D\big)$ | $\mathcal{O}\big(2T^2 N + TN^2\big)$ |
| Triplet-STS | $(1,2,0)$ | $\mathcal{O}\big((T^2 N + 2TN^2)D\big)$ | $\mathcal{O}\big(T^2 N + 2TN^2\big)$ |
| Quadruplet-TSST | $(2,2,0)$ | $\mathcal{O}\big((2T^2 N + 2TN^2)D\big)$ | $\mathcal{O}\big(2T^2 N + 2TN^2\big)$ |
| Quadruplet-STTS | $(2,2,0)$ | $\mathcal{O}\big((2T^2 N + 2TN^2)D\big)$ | $\mathcal{O}\big(2T^2 N + 2TN^2\big)$ |
| **Recurrent-based baselines (spatial grid $H \cdot W$)** | | | |
| ConvLSTM / PredRNN-family | – | $\mathcal{O}\big(T(HW)D^2\big)$ | $\mathcal{O}\big((HW)D\big)$ |
| **CNN-based baselines (spatial grid $H \cdot W$)** | | | |
| SimVP-family | – | $\mathcal{O}\big(T(HW)D^2\big)$ | $\mathcal{O}\big(T(HW)D\big)$ |

Let $c_t$ and $c_s$ denote the number of temporal and spatial self-attention blocks in one spatio-temporal unit (macro-block), respectively. For the Full Attention baseline, we denote by $c_j$ the number of joint spatio-temporal attention blocks. The per-unit complexity of a factorized variant is then given by:

$$\text{Compute} \sim \mathcal{O}\big((c_t T^2 N + c_s TN^2)D\big), \qquad \text{Memory} \sim \mathcal{O}\big(c_t T^2 N + c_s TN^2\big),$$

while for the Full Attention baseline it is

$$\text{Compute} \sim \mathcal{O}\big(c_j (TN)^2 D\big), \qquad \text{Memory} \sim \mathcal{O}\big(c_j (TN)^2\big).$$

Since the total depth is fixed, the overall complexity of the full network is linear in the number of units, and we omit this multiplicative factor for clarity. The resulting per-unit complexities for all nine PredFormer variants are summarized in Table 9.

**Normalization of attention blocks.** In our default configuration, we keep the number of *spatio-temporal units* the same for all variants, and each unit follows a specific ordering of temporal (T) and spatial (S) self-attention. For example, on Human3.6m, the Quadruplet-TSST variant is instantiated as `TSST`×3, which uses the same number of units as the Full Attention baseline; the only difference is that each unit in Quadruplet-TSST splits the attention into separate temporal and spatial operations, whereas the Full Attention baseline applies a joint self-attention over all $TN$ tokens. This design ensures that our factorized variants do not gain an unfair advantage by simply reducing depth; instead, the observed differences in FLOPs, memory, and empirical performance can be attributed to the factorization pattern itself.

## A.4 Experiment Setting

We provide our hyperparameter setting in Tab 10. For Moving MNIST, we use 24 GTB blocks for all PredFormer variants, which means 6 Quadruplet-TSST layers, 8 Triplet-TST layers, and 12 Binary-TS layers, respectively. For the TaxiBJ and WeatherBench datasets, we use 6 GTB blocks for the Triplet variants and 8 GTB blocks for the other variants.

Table 10: Hyperparameter Setting.

| Hyperparameter | Moving MNIST | TaxiBJ | WeatherBench | Human3.6m |
|---|---|---|---|---|
| **Training Hyperparameter** | | | | |
| Batch Size | 16 | 16 | 16 | 8 |
| Learning Rate | 1e-3 | 1e-3 | 5e-4 | {5e-4, 1e-3} |
| Learning Scheduler | Onecycle | Onecycle | Cosine | Cosine |
| Optimizer | Adamw | Adamw | Adamw | Adamw |
| Weight Decay | 1e-2 | 1e-2 | 1e-2 | 1e-2 |
| Training Epochs | 2000 | 200 | 50 | 50 |
| **Model Hyperparameter** | | | | |
| Patch Size | 8 | 4 | 4 | 8 |
| GTB Blocks | 24 | {6,8} | {6,8} | 12 |
| GTB Dim | 256 | 256 | 256 | 256 |
| GTB Heads | 8 | 8 | 8 | 8 |
| SwiGLU Hidden Dim | 1024 | 1024 | 512 | 1024 |
| Attention Dropout | 0.0 | 0.1 | 0.1 | 0.1 |
| SwiGLU Dropout | 0.0 | 0.1 | 0.1 | 0.1 |
| Drop Path Rate | 0.0 | 0.1 | 0.25 | 0.1 |

## A.5 More Experiments

### A.5.1 Comparison with Recent Recurrent Architectures

Tables 11 and 12 compare PredFormer with two recent recurrent architectures, SwinLSTM and VMRNN, on the Moving MNIST and TaxiBJ datasets, respectively. Across both benchmarks, PredFormer achieves lower MSE and higher SSIM while using comparable or fewer parameters and FLOPs. At the same time, PredFormer provides substantially faster training (shorter epoch time) and higher inference throughput (FPS), highlighting a favorable trade-off between accuracy and efficiency compared to these recurrent designs.

Table 11: Comparisons of PredFormer, SwinLSTM, and VMRNN on the Moving MNIST dataset.

| Method | Paras (M) | Flops (G) | Epoch Time | MSE | SSIM |
|---|---|---|---|---|---|
| SwinLSTM | 20.2 | 69.9 | 9min | 17.7 | 0.962 |
| VMRNN | – | – | 18min | 16.5 | 0.965 |
| PredFormer 3TSST Layer | 12.7 | 8.3 | **1.5min** | 16.2 | 0.965 |
| PredFormer 6TSST Layer | 25.3 | 16.5 | 3.5min | **12.5** | **0.973** |

Table 12: Comparison of PredFormer, SwinLSTM, and VMRNN on the TaxiBJ dataset.

| Method | Paras (M) | Flops (G) | Epoch Time | FPS | MSE | MAE | SSIM |
|---|---|---|---|---|---|---|---|
| SwinLSTM | 2.9 | 1.3 | – | 1425 | 0.303 | 15.0 | 0.9843 |
| VMRNN | **2.6** | **0.9** | 5min | 526 | 0.289 | 14.7 | 0.9858 |
| PredFormer | 6.3 | 1.6 | **1min** | **2354** | **0.277** | **14.3** | **0.9864** |

### A.5.2 Ablation on Patch Size

To further analyze the effect of patch size, we fix the architecture to the Triplet-TST variant of PredFormer and vary the patch size from 8 to 4 on Moving MNIST. As shown in Table 13, using a smaller patch size ($4 \times 4$) increases the number of spatial tokens from $N = 64$ to $N = 256$, which leads to higher computational cost (FLOPs 67.6G vs. 16.4G) and lower throughput (FPS 110 vs. 165). This finer granularity yields only a modest improvement in prediction accuracy (MSE 11.9 vs. 13.4, MAE 42.0 vs. 47.2, SSIM 0.974 vs. 0.971). Since the Triplet-TST variant with patch size 8 already outperforms all recurrent and convolutional baselines by a clear margin, we adopt patch size 8 as the default setting in the main experiments to achieve a better balance between accuracy and efficiency.

## A.6 More Visualizations

Fig 7(a) and (b) depict the inflow and outflow at the same time step. In this case, the fourth frame shows significantly less traffic flow than the previous frames. Constrained by the inductive bias of CNNs, TAU continues to predict high traffic levels while PredFormer demonstrates superior generalization by accurately

Table 13: Patch size ablation of PredFormer (Triplet-TST) on **Moving MNIST** after training for **2000** epochs. Each model observes 10 frames and predicts the subsequent 10 frames with an input resolution of $64 \times 64$.

| Patch Size | Resolution | Frames (in→out) | #Patches $N$ | Variant | Paras(M) | Flops(G) | FPS | MSE ↓ | MAE ↓ | SSIM ↑ |
|---|---|---|---|---|---|---|---|---|---|---|
| 4 | $64 \times 64$ | $10 \to 10$ | 256 | Triplet-TST | 25.3 | 67.6 | 110 | 11.9 | 42.0 | 0.974 |
| 8 | $64 \times 64$ | $10 \to 10$ | 64 | Triplet-TST | 25.3 | 16.4 | 165 | 13.4 | 47.2 | 0.971 |

Table 14: Dataset characteristics and optimal PredFormer variants. $P$ denotes patch size, $T$ is the input temporal length, $N$ is the number of spatial tokens after patch embedding, and $N/T$ measures the spatial–temporal token ratio. The "Best Variant(s)" and "Best MSE" are taken from the main results tables under the default hyperparameters.

| Dataset | Resolution $(H \times W)$ | Patch Size $P$ | #Patches $N$ | $T$ | $N/T$ Ratio | Best Variant(s) | Best MSE ↓ |
|---|---|---|---|---|---|---|---|
| Moving MNIST | $64 \times 64$ | 8 | $8 \times 8 = 64$ | 10 | 6 | Quadruplet-TSST, Quadruplet-STTS | 12.4 |
| Human3.6m | $256 \times 256$ | 8 | $32 \times 32 = 1024$ | 4 | 256 | Quadruplet-TSST | 110.9 |
| TaxiBJ | $32 \times 32$ | 4 | $8 \times 8 = 64$ | 4 | 16 | Binary-ST, Triplet-STS | 0.277 |
| WeatherBench | $32 \times 64$ | 4 | $8 \times 16 = 128$ | 12 | 11 | Fac-T-S | 1.100 |

capturing this abrupt change. This capability highlights PredFormer's potential to handle extreme cases, which could be particularly valuable in applications like traffic flow prediction and weather forecasting.

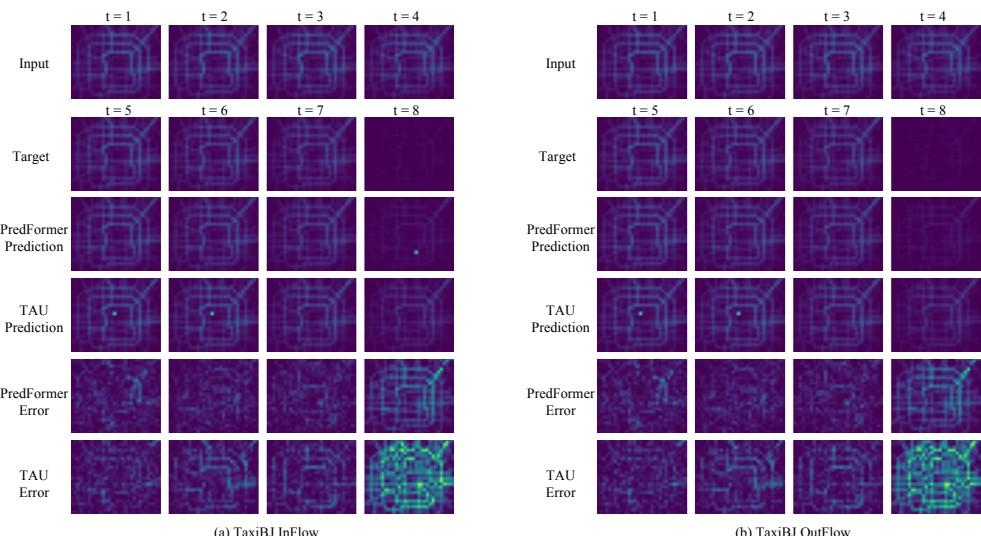

Figure 7: Visualizations on TaxiBJ InFlow and OutFlow. We amplify the error for better comparison.

## A.7 Empirical Discussion on Optimal PredFormer Variants Across Tasks

To better understand why different PredFormer variants become optimal on different benchmarks, we summarize in Table 14 the key spatio-temporal characteristics of each dataset together with the best-performing PredFormer variant(s). As shown in Table 14, all four benchmarks favor factorized variants of PredFormer rather than the full-attention baseline, but the specific optimal pattern is dataset-dependent:

On WeatherBench, the temporal horizon is the longest ($T = 12$) while the spatial fields are relatively smooth. In this setting, the Fac-T-S variant, which applies temporal attention before spatial attention, performs best, suggesting that emphasizing temporal modeling early is beneficial for long-range geophysical forecasting.

On Human3.6m, the temporal window is short ($T = 4$), but the spatial resolution is very high ($256 \times 256$ with $N/T = 256$), and human motion tightly couples spatial joints and temporal evolution. Here, the Quadruplet-

TSST variant, which interleaves two temporal and two spatial attention stages, works best, indicating that this dataset benefits from a balanced treatment of temporal and spatial dependencies.

On TaxiBJ, the temporal horizon is also short ($T = 4$), but the spatial grid is relatively low-resolution ($32{\times}32$, $N/T = 16$) and dominated by structured traffic-flow patterns. In this regime, variants that allocate more capacity to spatial modeling at the end of the block (Binary-ST and Triplet-STS) perform best, suggesting that refining spatial correlations is especially important.

On Moving MNIST, both quadruplet variants (TSST and STTS) achieve the best results under the default patch size $P = 8$, indicating that a balanced four-stage factorization is robust on this synthetic but relatively long-horizon ($T = 10$) benchmark.

Overall, these results indicate that different spatio-temporal regimes (temporal horizon, spatial resolution, and the $N/T$ ratio) naturally favor different factorization patterns within PredFormer, which explains why the optimal variant is not universal but dataset-specific.

### A.8 Theoretical Analysis on PredFormer Variants' Performance Differences

We consistently observe that TSST outperforms TS, which in turn outperforms TST on datasets such as Moving MNIST, TaxiBJ, and Human3.6M. An exception occurs in WeatherBench, where this trend diverges due to severe overfitting. To analyze this phenomenon, we examine the representational capacity of temporal-first interleaved models. Unlike prior work that performs spatial-temporal attention factorization with a shared MLP, our approach allocates a dedicated SwiGLU FFN to each spatial and temporal attention block, enhancing the model's learning capacity and expressiveness. Then, the PredFormer encoder can be viewed as a spatial-temporal transformer sequence (e.g., TSSTTSST). We propose that a key factor influencing model performance is the number of unique spatial-temporal subsequence (e.g., TS, TST, TSST) partitions enabled by a given sequence. Sequences with richer and more diverse partition patterns are better able to capture complex spatio-temporal dependencies. We formalize this intuition through a unique partition counting algorithm, as described in Fig 8, and report the corresponding statistics in Tab 15. Notably, the number of unique partitions correlates well with empirical performance across configurations, offering a plausible explanation for the effectiveness of TSST.

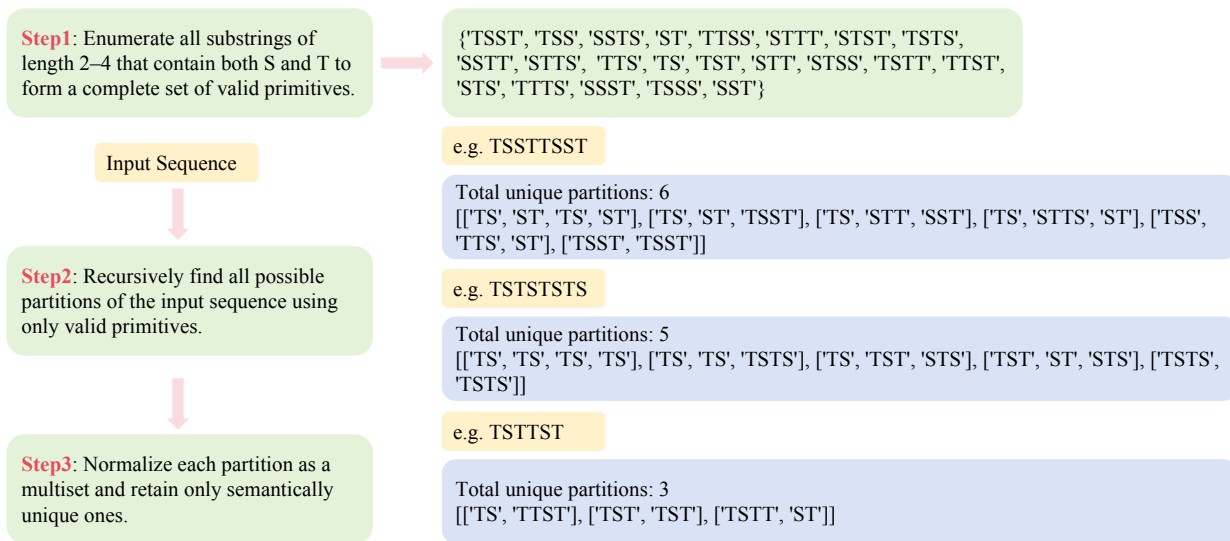

Figure 8: Spatial-Temporal GTB Unique Partition Algorithm.

Table 15: Analysis of Unique Partition Numbers.

| Moving MNIST | | | Human3.6m | | | TaxiBJ | | |
|---|---|---|---|---|---|---|---|---|
| Seq | Num ↑ | MSE ↓ | Seq | Num ↑ | MSE ↓ | Seq | Num ↑ | MSE ↓ |
| TSST*6 | **160** | **12.4** | TSST*3 | **16** | **110.9** | TSST*2 | 6 | 0.284 |
| TS*12 | 116 | 12.8 | TS*6 | 13 | 111.2 | TS*4 | 5 | **0.283** |
| TST*8 | 94 | 13.4 | TST*4 | 11 | 112.4 | TST*2 | 3 | 0.293 |

