# OpenReview forum: "Video Prediction Transformers without Recurrence or Convolution"
_TMLR — Accepted by TMLR_

### Review · Reviewer_Vzfr · 2025-11-10

**Summary Of Contributions:**

The paper proposes a spatio-temporal factorization variant of transformer architecture, called PredFormer, designed specifically for video prediction. The objective is to overcome the limitations of ViT (Vision Transformer)-integrated RNN and CNN-based approaches for spatio-temporal prediction. The results demonstrate both better efficiency than RNN-based approaches, and better prediction performance than CNN based approaches for 4 different datasets. The study also provides a technical report ablating some of the key design choices such as Gated Transformer block and use of learnt vs pre-defined positional embeddings. The main takeaway for me seems to be the idea that the temporal and spatial blocks should be factorized and processed separately by attention variant, and the interleaving between these blocks depends on the dataset properties. Though the novelty of this work is limited, specifically no new novel architecture has been proposed, the paper evaluates a simple idea and culminates in a nice technical report which might be of interest to some practitioners.

**Additional Comments:**

Not applicable.

**Audience:**

Yes

**Audience Explanation:**

As reflected in the summary, though the novelty of this work is limited, the paper evaluates the simple idea of spatio-temporal factorization in transformer attention and culminates in a nice technical report which might be of interest to some practitioners.

**Broader Impact Concerns:**

Not applicable.

**Claims And Evidence:**

Yes

**Claims Explanation:**

The paper provides various efficiency and fidelity quantitative results on four different datasets, comparing against various benchmarks validating their claims. Moreover, ablations are provided for various design decisions.

**Requested Changes:**

Overall there are a few weaknesses of this work. The choice of datasets, though diverse, is limited and has relatively small scale to make broader claims with respect to the benefits achievable by such architecture changes. Moreover, all chosen datasets are of extremely small resolution and temporal horizons. I would tone down conclusion statement 2 as some of these relative conclusions between the PredFormer variants may just be an artifact of the chosen datasets. I do not expect authors to scale up their experiments for securing my recommendation but a clarification on the dataset sizes should be included in the conclusion given the broader claim is on the architecture.

Second, I feel a qualitative analysis on why certain variants led to better performance on specific datasets could have been nice addition given the emphasis over variants in the paper. This can also clarify the benefits achieved from the spatio-temporal factorization of the architecture. Authors do take a half-pass attempt in Appendix A.5 to theoretically understand this but this section is clearly not a full-finished work and can be more fleshed out.

A few minor clarifications:
- Figure 4 has a typo: “(b) TaxiBJ”. It doesn’t show results for this data.
- Table 1: T, T’, interval definitions were unclear

---

### Review · Reviewer_cjnx · 2025-11-23

**Summary Of Contributions:**

The paper proposes PredFormer for video prediction. PredFormer utilizes a Gated Transformer Block to balance computational cost and performance for video prediction. Extensive experiments on four diverse benchmarks verify that PredFormer shows significant improvements in both accuracy and efficiency over strong baselines like SimVP and SwinLSTM.

Strengths:
1.The architecture of PredFormer is carefully designed. The paper provides a clear and systematic analysis of spatiotemporal factorization in transformers. The authors compare 9 distinct encoder architectures, including a baseline Full Attention, two standard factorized encoders (Fac-S-T, Fac-T-S), and six novel interleaved variants (Binary, Triplet, Quadruplet).

2.The empirical results are promising. The paper demonstrates state-of-the-art performance across four video prediction benchmarks. Moreover, the qualitative visualizations show better predictions and noticeably lower error residuals compared to the TAU baseline.

Weakness:
1.The goal of motivation and the method are mismatched. The introduction emphasizes that prior models “struggle to balance computational cost and performance” and that Transformers suffer from “quadratic scaling with sequence length.” However, the methods do not analyze how PredFormer’s architecture achieves computational efficiency. Moreover, the author does not discuss how the interleaved attention variants reduce cost.

2.The core component of PredFormer lacks novelty. The author presents GTB as a core component of PredFormer, which is widely used in former works [1-2]. However, the author does not discuss why this specific gating mechanism benefits spatiotemporal modeling in video prediction, which makes this component feel like a simple application rather than a novel contribution.

3.The spatial–temporal separation design is not a new concept, which is similar to recent model design [3] or time-series application [4-5]. I recommend that the author clarify what distinctive advantage this design offers in video prediction.

4.The claim of model variants lacks support. The author claims that “Triplet-TST captures more temporal dependencies, while Triplet-STS focuses more on spatial dependencies”. However, the paper does not provide any analytical or empirical evidence to support the claim.

5.The analysis on the compared baseline is incomplete and overstated.  The paper asserts that “using CNNs … limits the model’s performance ceiling,” but the experiments only compare with older CNN-based methods (e.g., SimVP, TAU, PredRNN). The author does not compare PredFormer with recent recurrent architectures such as RWKV-TS [6] or CNNs with large receptive fields (e.g., Wavelet Convolutions [7]).

6.The optimal variant on different tasks lacks explanation. Quadruplet-TSST is best for Moving MNIST, Triplet-STS is best for TaxiBJ, and Fac-T-S is best for WeatherBench. However, the paper gives no theoretical or empirical analysis of why this result occurs.

7.Lack of ablation on patch size. The author mentions splitting videos into non-overlapping 3D patches in Sec. 3.1. However, the author does not provide any ablation study to justify their choice or analyze its impact on performance and efficiency.

[1] Liu H, Dai Z, So D, et al. Pay attention to mlps[J]. Advances in neural information processing systems, 2021, 34: 9204-9215.

[2] Shazeer N. Glu variants improve transformer[J]. arXiv preprint arXiv:2002.05202, 2020.

[3] Tolstikhin I O, Houlsby N, Kolesnikov A, et al. Mlp-mixer: An all-mlp architecture for vision[J]. Advances in neural information processing systems, 2021, 34: 24261-24272.

[4] Ekambaram V, Jati A, Nguyen N, et al. Tsmixer: Lightweight mlp-mixer model for multivariate time series forecasting[C]//Proceedings of the 29th ACM SIGKDD conference on knowledge discovery and data mining. 2023: 459-469.

[5] Zhong Z, Yu Z, Yang Y, et al. PatchAD: A lightweight patch-based MLP-mixer for time series anomaly detection[J]. IEEE Transactions on Big Data, 2025.

[6] Hou H, Yu F R. Rwkv-ts: Beyond traditional recurrent neural network for time series tasks[J]. arXiv preprint arXiv:2401.09093, 2024.

[7] Finder S E, Amoyal R, Treister E, et al. Wavelet convolutions for large receptive fields[C]//European Conference on Computer Vision. Cham: Springer Nature Switzerland, 2024: 363-380.

**Audience:**

Yes

**Audience Explanation:**

1.The architecture of PredFormer is carefully designed. The paper provides a clear and systematic analysis of spatiotemporal factorization in transformers. The authors compare 9 distinct encoder architectures, including a baseline Full Attention, two standard factorized encoders (Fac-S-T, Fac-T-S), and six novel interleaved variants (Binary, Triplet, Quadruplet).

2.The empirical results are promising. The paper demonstrates state-of-the-art performance across four video prediction benchmarks. Moreover, the qualitative visualizations show better predictions and noticeably lower error residuals compared to the TAU baseline.

**Claims And Evidence:**

No

**Claims Explanation:**

1. The claim of model variants lacks support. The author claims that “Triplet-TST captures more temporal dependencies, while Triplet-STS focuses more on spatial dependencies”. However, the paper does not provide any analytical or empirical evidence to support the claim.

2. The analysis on the compared baseline is incomplete and overstated.  The paper asserts that “using CNNs … limits the model’s performance ceiling,” but the experiments only compare with older CNN-based methods (e.g., SimVP, TAU, PredRNN). The author does not compare PredFormer with recent recurrent architectures such as RWKV-TS [6] or CNNs with large receptive fields (e.g., Wavelet Convolutions [7]).

**Requested Changes:**

1. clarification of the motivation and method.

2. clarify the distinctive advantage of the method design in video prediction.

3. provide empirical evidence to suuport the claim "Triplet-TST captures more temporal dependencies, while Triplet-STS focuses more on spatial dependencies".

4. compare PredFormer with recent recurrent architectures such as RWKV-TS [6] or CNNs with large receptive fields (e.g., Wavelet Convolutions [7])

5. add explanation for the optimal variant on different tasks

6. add an ablation on patch size.

---

### Review · Reviewer_yZJo · 2025-11-27

**Summary Of Contributions:**

This paper proposes PredFormer, a pure transformer-based framework for video prediction that entirely eschews the use of recurrence and convolution, which are mainstays of prior work. The core idea is to leverage a simple encoder-decoder architecture where the encoder is composed of stacked "Gated Transformer Blocks" that process flattened video patches.

Contributions:
- introduction of PredFormer
- an exploration of nine different ways to model spatiotemporal dependencies using transformer blocks
- demonstration of state-of-the-art performance across four diverse and standard benchmarks (Moving MNIST, Human3.6m, TaxiBJ, WeatherBench). The proposed model not only achieves higher accuracy but does so with significantly improved computational efficiency (fewer parameters, lower FLOPs, and faster inference) compared to leading RNN- and CNN-based methods.

**Audience:**

Yes

**Audience Explanation:**

The claims of achieving state-of-the-art performance and superior efficiency seems well-supported, however I must disclose that I am not sure all the relevant baselines are present in the comparison as it's not directly my field.

**Claims And Evidence:**

Yes

**Claims Explanation:**

Key Strengths:

- Simplicity and Effectiveness: The paper argues for a simpler architectural paradigm for video prediction and backs it up with empirical results. The elegance of the pure transformer design is a major strength.
- Well-Written and Clear: The paper is easy to read and well-structured. The figures effectively illustrate the different architectures.

Potential Weaknesses:
- Incomplete Efficiency Analysis: The paper reports FLOPs and FPS but omits a discussion on memory requirements (e.g., peak GPU memory usage), I notably assume that KV cache has a huge memory footprint on deployment.
- Limited Theoretical Complexity Analysis: The computational complexity is primarily discussed through empirical metrics. A formal theoretical analysis of how the different variants scale with input dimensions would strengthen the paper's claims about efficiency.

**Requested Changes:**

1. Memory Usage Analysis (Necessary)
The current efficiency analysis focusing on FLOPs and FPS is good, however to provide a more complete picture, I suggest adding the peak GPU memory consumption during inference to the main results tables. This is a critical metric for practitioners evaluating models for real-world deployment, as I expect that the KV cache isn't light compared to RNN.

2. Theoretical Complexity Analysis (Necessary)
The paper would be significantly strengthened by a more formal, theoretical complexity analysis to complement the empirical results. Please consider adding a section or a table that provides the Big-O complexity of the main architectural variants for memory and compute. The analysis should be in terms of key parameters like sequence length (T), number of patches (N), and hidden dimension (D). This would provide clearer insight into how each variant scales and formalize the efficiency claims.

---

### Decision · Action_Editor_mzpw · 2026-01-02

**Recommendation:** Accept as is

**Audience:**

Yes

**Audience Explanation:**

Reviewers unanimously feel that this paper would be of interest to the machine learning community. The reviewers note that the paper is well and clearly written. The paper thoroughly evaluates the simple and elegant idea of spatio-temporal factorization in transformer attention. The SOTA performance on four video prediction benchmarks is encouraging for the usefulness of the proposed architecture. This is a good paper and should be accepted to TMLR.

**Claims And Evidence:**

Yes

**Claims Explanation:**

The paper presents a gated-transformer architecture for video prediction that avoids using recurrent or convolutional structures. The authors propose a simpler architectural paradigm for video prediction and support it with empirical evidence. The authors addressed the weaknesses pointed out by the reviewers in the original submission, including theoretical complexity analysis, additional baseline comparisons with more modern recurrent architectures, and ablations. The reviewers are in agreement that their concerns were addressed. Overall, the work presents a well-designed architecture with many supporting empirical results and analysis.